# Bond strength of thermoformed and 3D-printed aligners with universal primer versus one-step aligner adhesive with and without sandblasting: An in vitro study

**Viet Anh Nguyen**[ID][1]*, **Viet Hoang**[2], **Thi Quynh Trang Vuong**[3], **Thi Nga Phung**[3], **Nghi Phan Bich Hoang**[3]

**1** Department of Orthodontics, Faculty of Dentistry, Phenikaa University, Hanoi, Viet Nam, **2** Department of Orthodontics and Pedodontics, Faculty of Dentistry, Van Lang University, Ho Chi Minh City, Vietnam, **3** Private Practice, Viet Anh Orthodontic Clinic, Hanoi, Vietnam

* anh.nguyenviet1@phenikaa-uni.edu.vn

## Abstracts

### Objectives

Chairside bonding of auxiliaries directly to aligners can avoid remanufacturing trays, but optimal protocols may be substrate-specific across modern thermoformed and 3D-printed materials. This study aimed to compare bond strength and failure mode across six representative aligner materials using a universal primer-orthodontic adhesive combination and a one-step aligner adhesive, with and without sandblasting.

### Materials and methods

Polyethylene terephthalate glycol-modified (PETG), thermoplastic polyurethane (TPU), and glycol-modified polycyclohexylenedimethylene terephthalate (PCTG), together with three 3D-printed resins (TA-28, TC-85DAC, DCA), were prepared as 0.76-mm plates (n = 64). Specimens received alumina sandblasting or no treatment, then were bonded with either of two bonding strategies (n = 16). After thermocycling, bond strength was tested, and failures were scored by ARI. Two- and three-way ANOVA and proportional-odds modeling assessed effects (α = 0.05).

### Results

Bond strength showed significant main effects of material and sandblasting, with significant material–sandblasting and material–primer interactions. The primer main effect was not significant. Post hoc tests confirmed substrate-specific rankings. PETG with Bond Aligner (non-sandblasted) reached 26.71 MPa, while DCA with universal primer (sandblasted) reached 22.36 MPa. Sandblasting generally increased bond strength, with some exceptions. Failure mode was material-dependent and not completely parallel with bond strength.

**Data availability statement:** All relevant data are within the paper and its Supporting information files.

**Funding:** The author(s) received no specific funding for this work.

**Competing interests:** The authors have declared that no competing interests exist.

## Conclusions

Bonding efficacy depends on the aligner substrate. For thermoformed trays, a one-step aligner adhesive is preferable, with sandblasting contraindicated for PETG but advantageous for more elastic TPU and PCTG. For 3D-printed trays, a universal primer-orthodontic adhesive combination performs more consistently, with sandblasting benefiting DCA and TA-28, whereas TC-85DAC performs slightly better without it.

## 1. Introduction

Clear aligners have become a mainstream modality in contemporary orthodontics, largely because adults and image-conscious adolescents seek inconspicuous treatment options [1]. Their aesthetic appeal, removability, and the efficiencies of digital planning have positioned aligners as a first-line alternative to fixed appliances in mild-to-moderate malocclusions. Despite these advantages, many biomechanical tasks still require auxiliaries that are not intrinsic to the plastic tray. For inter-arch elastics, manufacturers generally provide precision cut-outs that expose enamel so a metal hook or button can be bonded directly to the tooth [2]. When the need for elastics emerges only after treatment has begun, because of unanticipated anchorage demands, late compliance issues, or refinements, ordering a new series of trays delays progress and increases cost. Clinicians sometimes punch their own cut-outs, yet uncontrolled trimming risks weakening the tray and altering its force delivery.

Direct bonding of hooks or buttons to the aligner itself has therefore gained popularity [3]. An in-vitro force-distribution study recently showed that bonded composite buttons transmitted roughly 96% of the applied elastic load and dispersed it across several teeth, whereas laser-cut hook traction lost more force and focused stresses chiefly on the canines, emphasizing the biomechanical advantage of directly bonding buttons to the aligner [4]. Beyond elastics, chairside bonding can facilitate anterior bite turbos or posterior bite blocks for open-bite correction or vertical control, applications where bulky virtual resin ramp can leave a large unsupported void beneath the tray and thereby compromise its fit. Functional or orthopedic adjuncts are also conceivable, such as mandibular-advancement devices or bondable pads of maxillary distalizers, which could be bonded directly to aligners [5].

Reliable bonding, however, is challenged by the diversity of emerging aligner substrates. Thermo-formable sheets range from polyethylene-terephthalate glycol (PETG) through thermoplastic polyurethane (TPU) to multilayer polycyclohexylenedimethylene-terephthalate glycol-modified (PCTG), while additive-manufactured trays are now printed from photopolymers such as urethane dimethacrylate resins [6,7]. Available primers likewise vary, from universal 10-MDP adhesives to proprietary resin blends specifically formulated for aligner materials. To our knowledge, published evidence on bonding to aligner plastics is virtually confined to several studies [8,9]. Pariyatdulapak et al. reported highly variable bond strengths when metal-mesh buttons were bonded to PETG and TPU foils with one paste adhesive. However, the button mesh conferred additional mechanical retention, and the test method did not follow ISO 29022:2013, limiting the relevance of its

findings to direct composite-to-aligner bonding [10]. Another technical report on attaching buttons directly onto clear aligners mainly describes a chairside method for creating elastic anchors, without reporting standardized shear bond strength or characterizing the composite-to-aligner interface [9].

Against this backdrop, the present in-vitro investigation compared the shear-bond performance of two clinically popular adhesive systems on six representative aligner materials, three thermo-formed foils (PETG, TPU, and PCTG-based tri-layer) and three 3D-printed resins, under standardized surface preparations and thermal aging. The findings aim to guide evidence-based selection of bonding protocols when mid-treatment auxiliaries must be added without remanufacturing the entire aligner series.

## 2. Materials and methods

### 2.1. Study design

This cross-sectional in-vitro laboratory study involved no human participants or identifiable data and therefore did not require institutional review board approval. The study was reported in accordance with the modified CONSORT checklist for in vitro studies of dental materials to enhance transparency and reproducibility [11]. An a priori power analysis was performed with G*Power 3.1 (Heinrich-Heine University, Düsseldorf, Germany) for an ANOVA with fixed effects, special, main effects, and interactions. With an assumed medium effect size ($f = 0.25$), an alpha level of 0.05, a desired statistical power of 0.95, and five numerator degrees of freedom, the calculation yielded a total required sample size of 323, corresponding to a minimum of 14 specimens for each experimental cell. Accordingly, sixteen specimens were prepared for each cell to allow for potential losses.

### 2.2. Aligner materials and surface treatment

Six commercially available aligner resins/foils were evaluated (Table 1). Thermo-formable sheets of PETG, TPU, and copolyester (PCTG) were supplied as 0.76 mm foils (Shanghai Maxflex Medical Technology, Shanghai, China). Additionally, three-dimensional (3D) printing materials, including TA-28 (Graphy, Seoul, South Korea), TC-85DAC (Graphy) and DCA (LuxCreo, Beijing, China), were printed as 0.76-mm flat plates (20 mm x 10 mm) positioned flush against the build-platform membrane and post-cured following each manufacturer's instructions. All sheets were sectioned into 68 specimens with a 20-mm length and 10-mm width.

For each aligner material, 68 specimens were divided equally into sandblasted and non-sandblasted subgroups ($n = 34$). Sandblasting was carried out with 110-µm $Al_2O_3$ for 10 s at 0.30 MPa and a 10 mm stand-off distance using an AX-B5 sandblaster (IRIS, Tianjin, China). Sandblasting was selected as the sole surface treatment because it is widely applied in dental bonding, whereas alternative methods such as laser texturing, plasma, or chemical etchants are not routine for clear aligners and may not be approved for intraoral use on proprietary tray materials [12].

All samples were then cleaned with 96% isopropyl alcohol and air-dried. Two specimens from each subgroup were examined under a field-emission scanning electron microscope (SEM) (Quanta 450 FEG, Thermo Fisher Scientific, Hillsboro, OR, USA) at 20 kV to document surface morphology. The remaining thirty-two specimens were assessed for arithmetic mean roughness (Ra) with a contact profilometer (SRT-6200, Landtek, Guangzhou, China) using a 2.5 mm cut-off length.

### 2.3. Bonding protocol

Within each surface-treatment subgroup, sixteen specimens were bonded with a two-step universal system and sixteen with a one-step aligner system. For the universal protocol, Single Bond Universal primer (3M ESPE, St Paul, MN, USA) was actively rubbed on the aligner surface for 10 sec and gently air-thinned for 5 sec. A cylindrical silicone mold (internal diameter 3 mm, height 3 mm) was then packed with GoTo orthodontic composite (Reliance Orthodontic Products, Itasca, IL, USA) and positioned perpendicular to the primed surface. Excess material was removed with a micro-brush, and the

**Table 1. Commercial aligner substrates and bonding agents evaluated in this study, with their manufacturer and principal chemical composition.**

| Commercial name | Type | Manufacturer | Compositions | Elastic modulus (Mpa) |
|---|---|---|---|---|
| Max Economical | Thermo-formed foils | Shanghai Maxflex Medical Technology, Shanghai, China | Polyethylene-terephthalate-glycol (PETG) | 2310 |
| Flex Comfort | Thermo-formed foils | Shanghai Maxflex Medical Technology, Shanghai, China | Polyurethane (TPU) | 1450 |
| Flex Premium | Thermo-formed foils | Shanghai Maxflex Medical Technology, Shanghai, China | Polycyclohexylenedimethylene-terephthalate glycol-modified (tri-layer: PCTG-TPU-PCTG) | 1050 |
| Tera Harz TA-28 | Directly 3D printed resin | Graphy, Seoul, South Korea | Urethane dimethacrylate-based photopolymer | 1849 |
| Tera Harz TC-85DAC (Directly 3D printed) | Directly 3D printed resin | Graphy, Seoul, South Korea | Urethane dimethacrylate-based photopolymer | 1575 |
| Digital Clear Aligner DCA (Directly 3D printed) | Directly 3D printed resin | LuxCreo, Beijing, China | Bis-acrylate photopolymer | 1219 |
| Single Bond Universal | Universal primer | 3M ESPE, St Paul, MN, USA | Dimethacrylate resins, 2-hydroxyethyl methacrylate, vitrebond copolymer, 10-methacryloyloxydecyl dihydrogen phosphate, filler, ethanol, water, silane | |
| GoTo | Orthodontic adhesive | Reliance Orthodontic Products, Itasca, IL, USA | Bisphenol A-glycidyl methacrylate, glass filler, ethoxylated bisphenol A dimethacrylate, triethylene glycol dimethacrylate | 14200 |
| Bond Aligner | Aligner primer and adhesive | Reliance Orthodontic Products, Itasca, IL, USA | Tricyclodecane dimethanol diacrylate, N,N-dimethylacrylamide, pentaerythritol tetrakis 3-mercaptopropionate, phenyl bis 2,4,6-trimethylbenzoyl phosphine oxide | |

assembly was light-cured for 30 sec with a curing unit (LedF, Woodpecker, Guilin, China) delivering 1200 mW cm$^{-2}$. For the one-step protocol, Bond Aligner paste (Reliance Orthodontic Products, Itasca, IL, USA) was injected directly into the same mold, placed on the unprimed surface, excess material removed, and photo-polymerized under identical light-curing conditions. The silicone jig was then gently detached, leaving a composite cylinder suitable for subsequent shear testing.

## 2.4. Artificial aging

All bonded specimens were thermocycled 500 times between 5 °C and 55 °C in a dual-bath thermocycler (YTST-021, Yuanyao, Guangdong, China) using a 25-sec dwell period and a 10-sec transfer time. The cycle count was selected to approximate 1–2 weeks of intra-oral service [13]. Because the widely accepted conversion proposed by Gale et al., about 10,000 cycles per clinical year, 500 cycles represent the thermal load experienced over the maximum recommended aligner-wear interval of 14 days [14].

## 2.5. Shear bond strength testing and failure-mode analysis

After thermocycling, each specimen was bonded to a 5 mm-thick industrial acrylic plate (Shinkolite, Mitsubishi, Tokyo, Japan) using X2000 cyanoacrylate adhesive (White Tiger, Bangkok, Thailand) to provide a rigid backing and prevent flexure during shear-bond testing. Each specimen was then clamped directly in the grips of a universal testing machine (HP-1 kN; Handpi Instruments, Shenzhen, China) with the bonded interface oriented parallel to the loading blade. A chisel-shaped stainless-steel knife-edge was advanced at 1 mm/min until debonding occurred (Fig 1). The peak load was recorded in Newtons and divided by the bonded area to obtain shear-bond strength in megapascals (MPa).

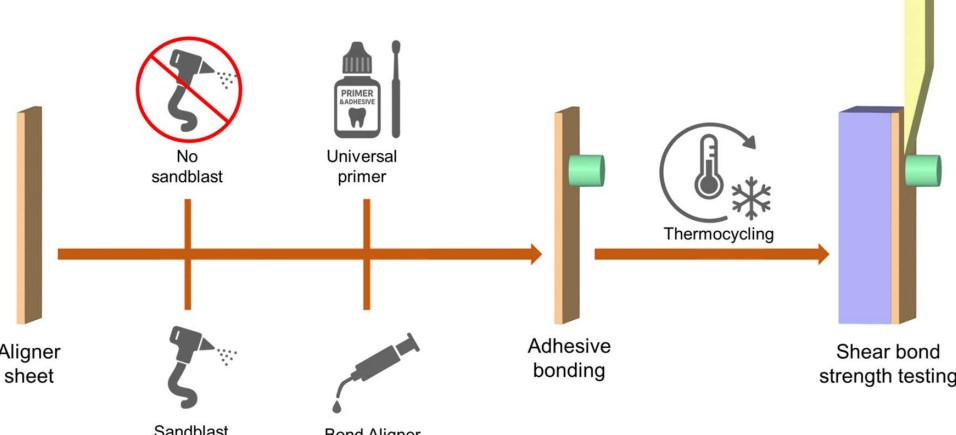

**Fig 1. Study flowchart.** Each material was split into two surface-treatment conditions: no treatment and sandblasting. Within each surface condition, specimens were bonded using either a two-step universal primer-orthodontic adhesive combination or a one-step aligner adhesive. After thermocycling, all bonded specimens underwent shear bond strength testing.

Debonded surfaces were examined under a 5x loupe and classified with the Adhesive Remnant Index (ARI), which assigns scores from 0 to 3 [15]. A score of 0 indicates that no adhesive remains on the aligner and therefore represents adhesive failure. A score of 1 corresponds to less than 50 per cent of the adhesive remaining, interpreted as a mixed failure that is predominantly adhesive. A score of 2 corresponds to more than 50 per cent adhesive remaining, interpreted as a mixed failure that is predominantly cohesive. A score of 3 indicates that the adhesive layer remains completely on the aligner, signifying cohesive failure either within the adhesive itself or within the aligner substrate.

### 2.6. Statistical analysis

Continuous outcomes, including surface roughness and shear bond strength, were summarized as mean ± standard deviation. Ra was analyzed with a two-way ANOVA (material × sandblasting). Multiple comparisons used Tukey HSD among materials within each sandblasting level, and within-material sandblasted vs non-sandblasted contrasts used Welch's t-test with Holm adjustment. Bond strength was analyzed with a three-way ANOVA (material × sandblasting × primer), followed by Tukey honestly significant difference (HSD) tests. The ordinal failure mode was evaluated using a proportional-odds ordinal logistic model including all main effects and interactions, with likelihood-ratio tests for effect blocks and predicted probabilities tabulated per material x sandblasting x primer. Statistical significance was set at α = 0.05.

### 3. Results

On surface roughness, two-way ANOVA identified significant main effects of material (F = 135.0, p < 0.001) and sandblasting (F = 2343.0, p < 0.001), as well as a significant material x sandblasting interaction (F = 88.9, p < 0.001). Across all materials, sandblasting produced higher Ra than the corresponding non-sandblasted condition (Welch's t tests with Holm correction, all p < 0.001). Within the non-sandblasted condition, PCTG exhibited the lowest Ra, TPU and DCA formed the lowest statistical group, PETG was intermediate, and TA-28 and TC-85DAC were higher (Table 2). Within the sandblasted condition, TA-28 had the highest Ra, followed by PETG and PCTG. TC-85DAC overlapped statistically with PCTG and DCA, whereas TPU remained the lowest.

Regarding SEM analysis, baseline micrographs showed smooth, feature-poor surfaces across all materials, including the thermoformed and 3D-printed sheets (Fig 2). After sandblasting, every thermoformed group exhibited pronounced

**Table 2. Surface roughness (Ra) by aligner material and sandblasting.**

| Material | No sandblast | Sandblast |
|---|---|---|
| PETG | 0.119 ± 0.042[ab,α] | 1.505 ± 0.193[a,β] |
| TPU | 0.132 ± 0.021[a,α] | 0.346 ± 0.082[b,β] |
| PCTG | 0.079 ± 0.010[b,α] | 1.222 ± 0.246[c,β] |
| TA-28 | 0.376 ± 0.113[c,α] | 1.828 ± 0.283[d,β] |
| TC-85DAC | 0.343 ± 0.108[c,α] | 1.108 ± 0.450[ce,β] |
| DCA | 0.150 ± 0.065[a,α] | 0.995 ± 0.206[e,β] |

Values are mean ± SD (µm). Superscript letters (a-e) denote column-wise multiple comparison (Tukey HSD, α = 0.05); groups sharing a letter are not significantly different. The superscript (α, β) denotes a significant row-wise difference (Welch's t-test with Holm correction, p < 0.05).

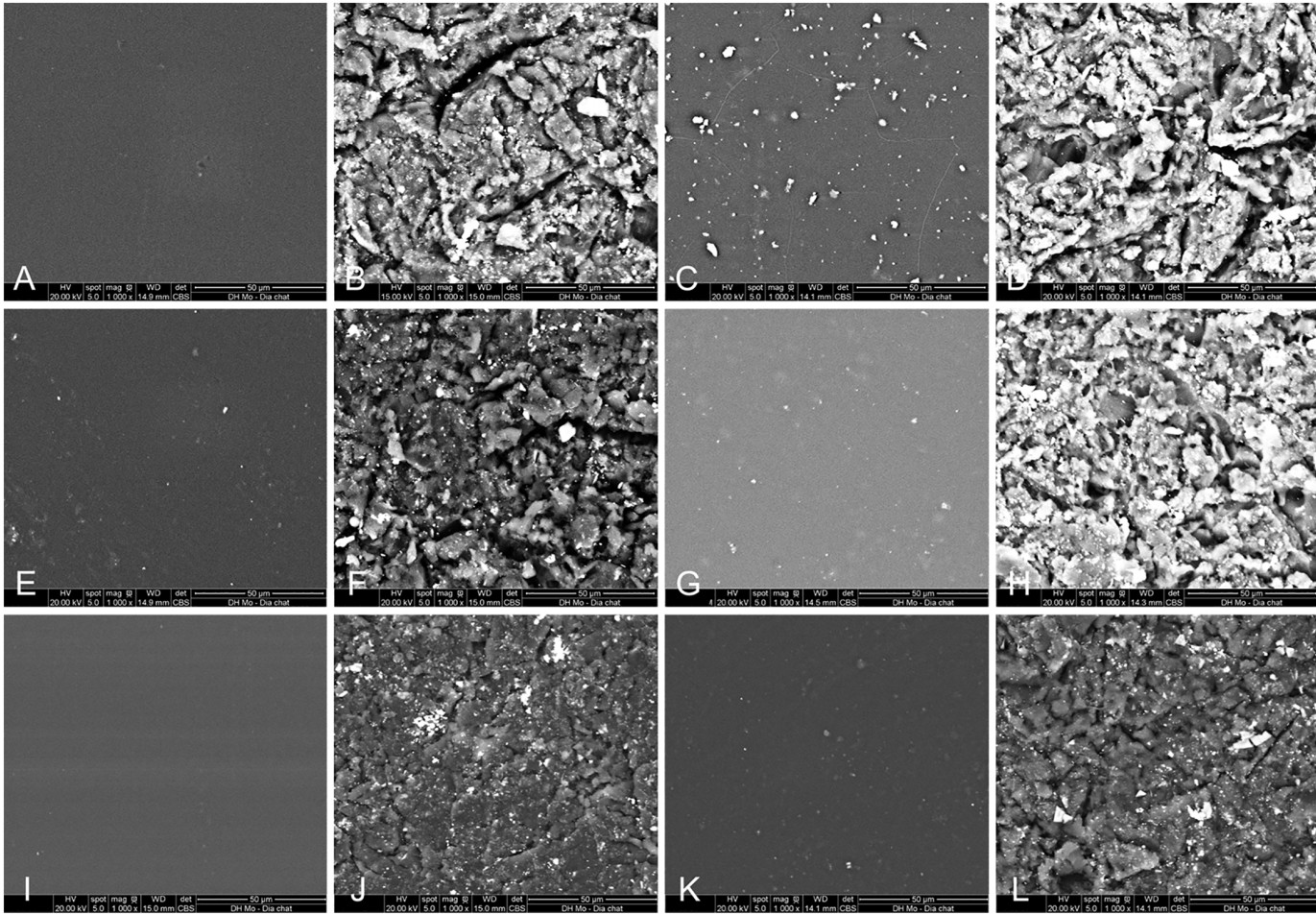

**Fig 2. Scanning electron micrographs (×1000) of aligner surfaces before (left panel) and after sandblasting (right panel). A,B,** PETG; **C,D,** TPU; **E,F,** PCTG; **G,H,** TA-28; **I,J,** TC-85DAC; **K,L,** DCA. Thermoformed foils and TA-28 showed pronounced ploughing, pits, and sharp asperities after sand-blasting. TC-85DAC and DCA showed comparatively shallower micro-cracking and less crater formation.

brittle ploughing with angular pits and sharp asperities, and the printed TA-28 splint developed a similar ploughed, cratered topography. In contrast, the printed TC-85DAC and DCA surfaces remained comparatively flat with fewer irregularities, characterized mainly by shallow, discontinuous surface cracks rather than extensive crater formation.

On shear bond strength, the three-way ANOVA showed significant main effects of material (F = 101.32, p < 0.001) and sandblasting (F = 54.59, p < 0.001), whereas primer alone was not significant (F = 0.46, p = 0.499). Two-way interactions were significant for material x sandblasting (F = 13.23, p < 0.001) and material x primer (F = 51.41, p < 0.001), and the three-way interaction was also significant (F = 9.99, p < 0.001), indicating that the effects of surface treatment and primer varied by tray material (Table 3).

Post-hoc Tukey groupings were consistent with these interactions (Table 4). Within the non-sandblasted, universal-primer condition, TC-85DAC and DCA formed the highest group, followed by PETG and TA-28, with TPU and PCTG lowest (Fig 3). Under the non-sandblasted, Bond-Aligner condition, PETG was highest, followed by DCA; both exceeded the other materials. With the sandblasted, universal-primer condition, DCA was highest, TC-85DAC next, PETG and TA-28 intermediate, and PCTG and TPU lowest. For the sandblasted, Bond-Aligner condition, PETG and DCA formed the highest group, TA-28 was lowest, and TPU, PCTG, and TC-85DAC were intermediate with overlap. Row-wise Tukey tests showed material-dependent condition effects. PETG and PCTG displayed stepwise increases from universal-primer to Bond-Aligner and with sandblasting.

Among the 3D-printed aligner materials, within each sandblast-condition subgroup, the universal primer generally exceeded Bond Aligner. The primer difference was significant for TC-85DAC in both subgroups (no sandblast p < 0.001, sandblast p = 0.002), significant for DCA only in the sandblasted subgroup (p = 0.009). Sandblasting tended to increase bond strength relative to the corresponding non-sandblasted condition, except for PETG with Bond Aligner (p < 0.001) and TC-85DAC with universal primer (p = 0.462). The sandblasting-non-sandblasting contrast was significant for TPU with

**Table 3. Three-way ANOVA for shear bond strength (MPa).**

| Effect | df | Mean square | F | p |
|---|---|---|---|---|
| Material | 5.0 | 1159.665 | 101.32 | <0.001 |
| Sandblast | 1.0 | 624.827 | 54.59 | <0.001 |
| Primer | 1.0 | 5.244 | 0.46 | 0.499 |
| Material x sandblast | 5.0 | 151.392 | 13.23 | <0.001 |
| Material x primer | 5.0 | 588.422 | 51.41 | <0.001 |
| Sandblast x primer | 1.0 | 1.6 | 0.14 | 0.709 |
| Material x sandblast x primer | 5.0 | 114.322 | 9.99 | <0.001 |

**Table 4. Shear bond strength by material and sandblasting-primer combination.**

| Material | No sandblast | | Sandblast | |
|---|---|---|---|---|
| | Universal primer | Bond aligner | Universal primer | Bond aligner |
| PETG | 12.10 ± 4.07[b,α] | 26.71 ± 3.35[a,β] | 14.02 ± 4.75[bc,α] | 19.38 ± 4.56[c,γ] |
| TPU | 7.89 ± 2.38[a,α] | 5.31 ± 1.48[c,β] | 9.04 ± 1.66[a,αγ] | 10.83 ± 2.02[ab,γ] |
| PCTG | 4.85 ± 1.53[a,α] | 8.95 ± 5.66[c,β] | 9.87 ± 2.93[a,β] | 14.13 ± 4.48[b,γ] |
| TA-28 | 9.72 ± 2.06[b,αβ] | 7.05 ± 4.44[c,α] | 11.40 ± 2.29[ab,β] | 10.51 ± 2.32[a,β] |
| TC-85DAC | 16.95 ± 3.76[c,α] | 6.78 ± 2.19[c,β] | 15.22 ± 3.53[c,α] | 10.86 ± 3.56[ab,γ] |
| DCA | 15.88 ± 4.35[c,αβ] | 13.22 ± 3.45[b,α] | 22.36 ± 2.46[d,γ] | 18.39 ± 3.23[c,β] |

Values are mean ± SD (MPa). Superscript letters (a-d) denote Tukey HSD groupings across materials within the same column; groups sharing a letter are not significantly different at α = 0.05. Superscript letters (α-γ) denote Tukey HSD groupings across the four conditions within the same row.

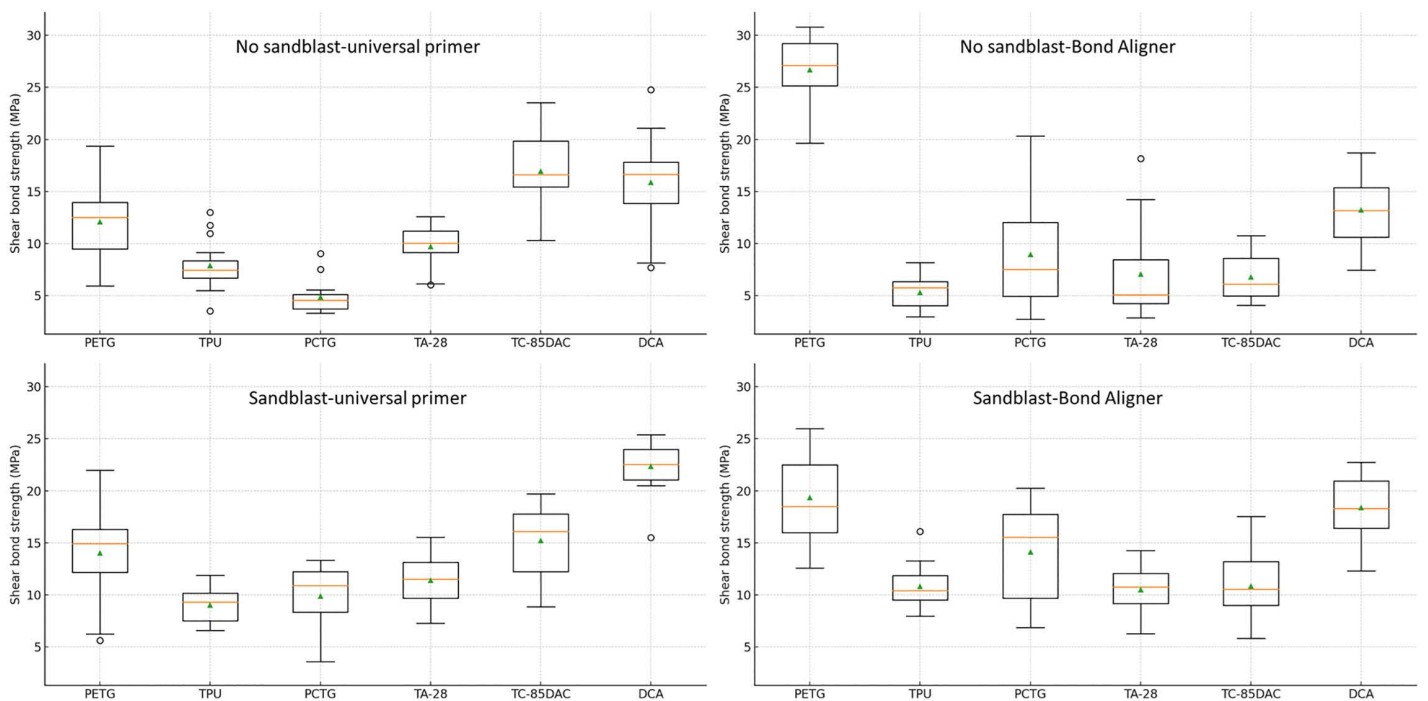

**Fig 3. Box plots of shear bond strength (MPa) by material, surface treatment (non-sandblasted versus sandblasted), and primer (universal primer–orthodontic adhesive combination versus one-step aligner adhesive).**

Bond Aligner (p < 0.001), PCTG under both primers (Bond Aligner p = 0.003, universal p = 0.004), TA-28 with Bond Aligner (p = 0.008), TC-85DAC with Bond Aligner (p = 0.005), and DCA under both primers (p < 0.001).

On failure mode, the proportional-odds ordinal logistic showed a significant material main effect ($\chi^2 = 92.35$, p < 0.001), whereas sandblasting ($\chi^2 = 0.18$, p = 0.674) and primer ($\chi^2 = 0.31$, p = 0.575) were not significant. Likelihood-ratio tests further indicated a significant material x primer interaction ($\chi^2 = 86.39$, p < 0.001) and a significant three-way interaction ($\chi^2 = 33.09$, p < 0.001), while material x sandblasting ($\chi^2 = 9.13$, p = 0.104) and sandblasting x primer ($\chi^2 \approx 0.00$, p = 0.989) were not significant. The highest proportion of ARI 3 occurred in PETG under the non-sandblasted-Bond-Aligner condition (50.0%). ARI 0 was observed entirely in PCTG-non-sandblasted subgroups for both primers. Across all sandblasting-primer combinations, PCTG consistently skewed toward lower ARI levels (Fig 4). In mixed failures across all material-primer-sandblasting combinations, cohesive damage occurred predominantly within the tray substrates rather than within the adhesive (Fig 5).

## 4. Discussion

This study evaluated the influence of aligner material, primer, and sandblasting on shear-bond strength and failure mode in a standardized in vitro model. The a priori null that shear-bond strength and failure mode would not differ by aligner material, primer, or surface treatment was only partly supported. Material and sandblasting showed significant main effects, whereas the primer main effect was not significant. Significant interactions indicate that primer efficacy is substrate dependent rather than uniformly null. We therefore reject the null for material and for sandblasting and reject a simplified claim that an aligner-specific adhesive consistently outperforms a universal primer across substrates. Failure mode distributions are also clustered by material, underscoring substrate-driven interfacial behavior.

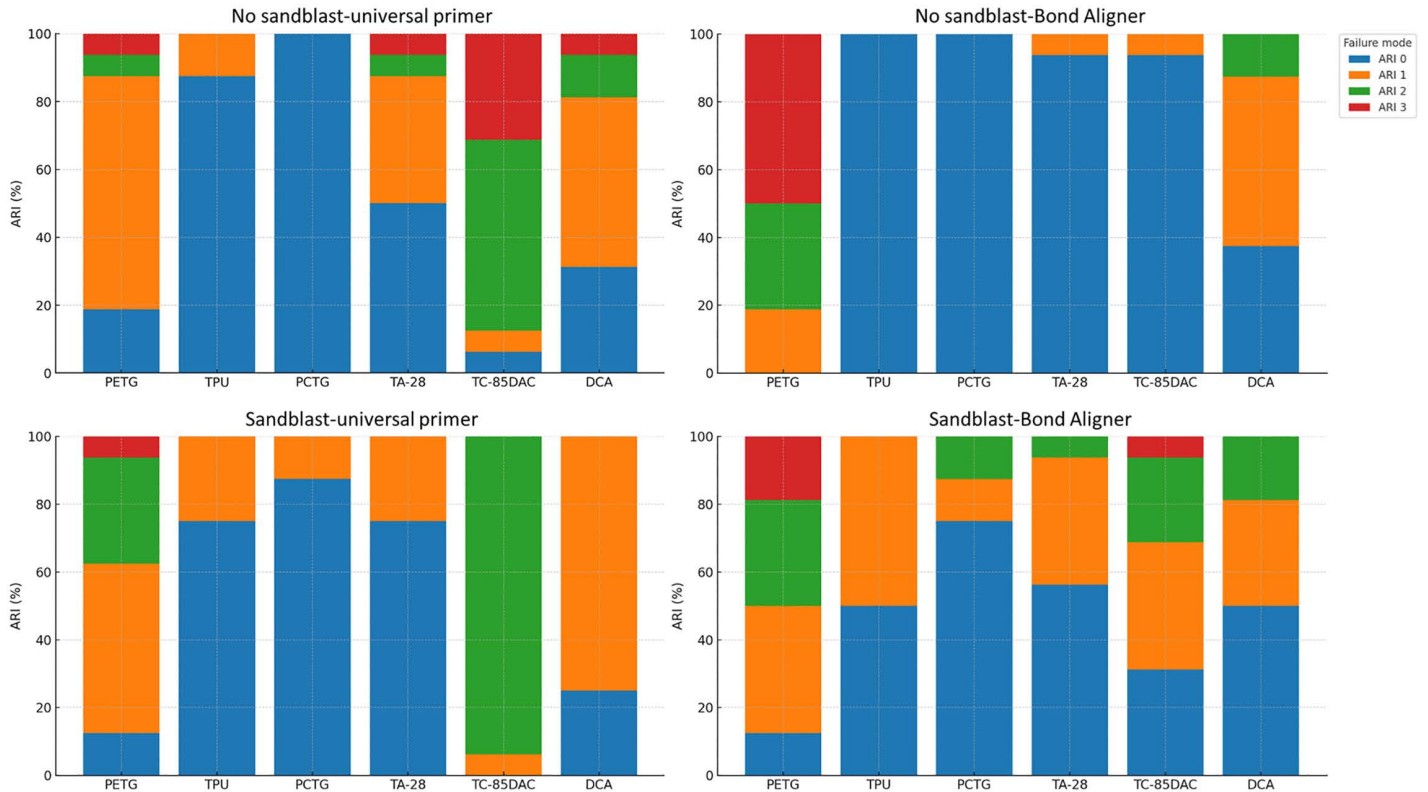

**Fig 4. Distribution of failure modes with Adhesive Remnant Index (ARI) across combinations of material, primer, and surface treatment.** PETG bonded with Bond Aligner tended to show higher ARI 3, whereas PCTG showed mostly ARI 0.

Consistent with the roughness outcomes, SEM findings were largely concordant. Before sandblasting, all materials showed smooth, feature-poor surfaces with low Ra values. After sandblasting, the pronounced brittle ploughing with angular pits and sharp asperities of the thermoformed sheets is consistent with the observed rise in Ra. Despite a cratered SEM appearance, the low Ra for TPU on contact profilometry likely reflects viscoelastic compliance and tip-convolution, which flatten peaks and undersample narrow, shallow pits [16]. Among the 3D-printed sheets, TA-28 showed greater post-sandblasting topographic disruption, whereas TC-85 and DAC changed less, corresponding to their smaller increases in Ra.

The higher bond strength of Bond Aligner on PETG and PCTG but not on TPU likely reflects a closer chemical and mechanical match between its ultraviolet curable acrylate and thiol network and the surface chemistry and mechanics of the polymer sheets. The formulation contains tricyclodecane dimethanol diacrylate, N,N-dimethylacrylamide, pentaerythritol tetrakis(3-mercaptopropionate), and a phosphine oxide photoinitiator, which interact favorably with polar copolyester surfaces, whereas TPU is a more compliant polyurethane with distinct surface energetics [17,18]. PETG and PCTG are copolyesters that include 1,4-cyclohexanedimethanol (CHDM) and ethylene glycol, and PCTG often has a higher CHDM fraction that yields greater ductility with modestly lower stiffness compared with many PETG grades [6]. Published reports indicate that PETG and PCTG have a higher elastic modulus than TPU, and in our study, the PCTG tray was a three-layer laminate with a TPU core [19,20]. Greater compliance in TPU and in softer tri-layer PCTG likely increases local bending and peel during shear testing, which penalizes a rigid glassy adhesive, whereas the stiffer PETG sheet limits flexure and helps sustain interfacial shear, consistent with the highest strengths observed for PETG across sandblast and primer conditions.

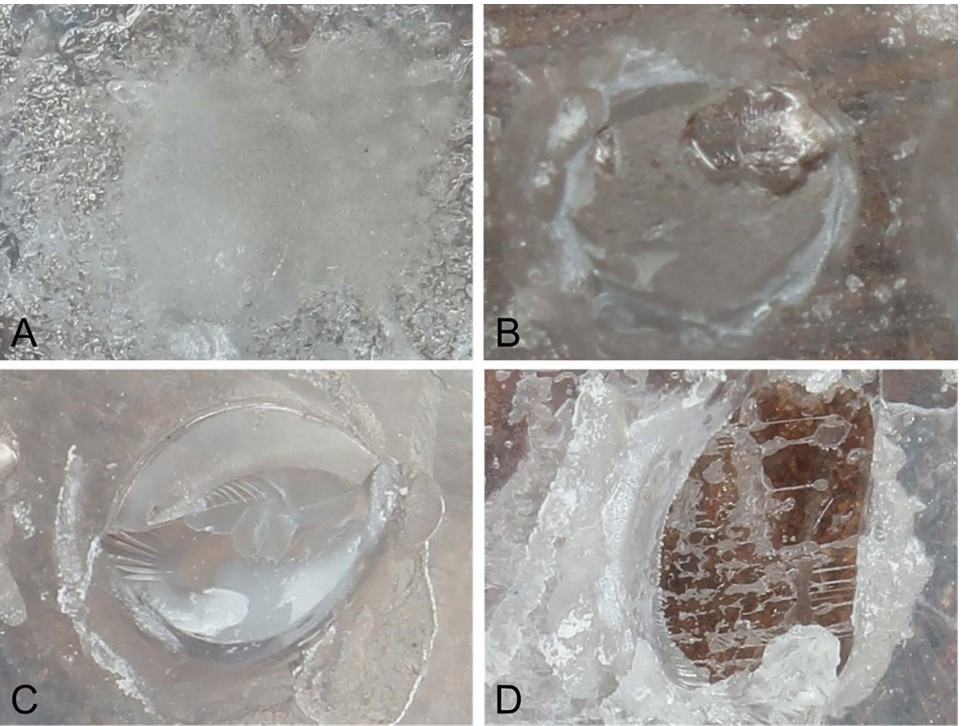

**Fig 5. Representative failure modes as per Adhesive Remnant Index (ARI). A,** ARI 0, adhesive fully on the composite cylinder; **B,** ARI 1, mixed failure with <50% adhesive left on tray; **C,** ARI 2, mixed failure with >50% adhesive left on tray; **D,** ARI 3, cohesive failure within the adhesive or tray polymer.

In contrast, the higher bond strength of universal primer on the 3D-printed aligners compared to Bond Aligner may be explained by chemical compatibility between the dimethacrylate-based resin matrix of the primer and the photopolymerizable resins used for these aligners. All three printed materials are methacrylate-based photopolymers. The methacrylate functional groups in the universal primer can co-polymerize with unreacted double bonds remaining on the surface of the printed resin after post-curing, creating covalent links across the interface in addition to micromechanical retention. Accordingly, bond strength likely depends on the availability of residual surface double bonds that can copolymerize. For TC-85DAC, in-vitro studies reported post-curing resulting in 96.5–98.6% conversion in nitrogen [21,22]. Comparable surface conversion data for TA-28 and DCA are not yet published. However, manufacturer guidance specifies nitrogen curing for TA-28 and TC-85 to mitigate oxygen inhibition, whereas the post-processing workflow for DCA does not require nitrogen-purged curing [23]. These process differences, together with oxygen-inhibition effects at air-exposed surfaces, could contribute to the lower bond strength observed for TA-28 and the trend toward higher values for DCA [24].

Sandblasting generally enhances adhesion by increasing surface roughness and micromechanical interlocking, which explains the overall upward shift with sandblasting [12]. The PETG-Bond Aligner exception is consistent with an adhesive that relies on strong chemical affinity to copolyesters and is applied without a separate primer step, so added roughness can impair wetting and intimate contact, and residual abrasive can counteract any mechanical gain. For TC-85DAC with the universal primer, the negligible change is compatible with a competing mechanism in which aggressive abrasion can remove unreacted double bonds for chemical coupling, offsetting the roughness benefit.

Notably, the PETG-non-sandblasted-Bond Aligner combination showed the largest ARI 3 proportion and the highest bond strength, consistent with stronger adhesion at the aligner-adhesive interface. Across the remaining groups, the

relation between bond strength and ARI was not monotonic, likely because failure mode also depends on the cohesive shear capacity of the aligner substrate. The predominance of mixed failures with tearing within the tray indicates that once interfacial adhesion surpasses the tray's cohesive strength, fracture propagates through the polymer sheet rather than through the adhesive, yielding variable ARI distributions even at comparable bond strengths.

Compared with the sole prior report of Pariyatdulapak et al. [8], which bonded orthodontic buttons with mesh bases to aligner materials using Bond Aligner and found 4.63–7.04 MPa (PETG) and 1.17–1.40 MPa (TPU), with PETG failures dominated by substrate tear-out, we observed markedly higher strengths (26.71 MPa on PETG and 5.31 MPa on TPU). The discrepancy likely reflects methodology, in which the earlier button–sheet joint wasn't bonded to a rigid industrial-plastic backing. Therefore, the sheet bent under load during testing, facilitating peel and easier debonding, whereas our direct composite-to-aligner protocol, with rigid support, isolates true adhesive performance. Furthermore, the peak values are still within the general magnitude recently reported for composite attachments bonded to other materials (12–15 MPa), suggesting that the present results are biomechanically plausible rather than artefactually inflated [25–27].

Clinically, these findings should be interpreted with caution. Our specimens were supported by a rigid backing to suppress tray flexion, whereas aligners in vivo bend under load, introducing peel components that can reduce effective shear capacity. In addition, intraoral moisture and continuous water exposure may further weaken the interface over time, which was not replicated in the present setup. Therefore, although most groups exceeded the commonly cited clinical threshold of approximately 6–8 MPa, real-world performance may be lower, suggesting in vivo validation [28,29]. It is also notable that several protocol–material combinations produced bond strengths that exceeded what is generally considered clinically sufficient. In practice, once retention is 'strong enough' to withstand routine elastic wear for the lifespan of a tray, additional strength may not translate into added clinical benefit. However, if debond does occur under chewing, the failure is more likely to involve local tearing of the tray rather than enamel damage. This suggests that very high bond strength should not be interpreted as a clinical liability.

From a clinical standpoint, the study results suggest a decision tree rather than a 'one size fits all' protocol. For thermoformed aligners, Bond Aligner is preferable to a universal primer. Sandblasting is contraindicated for PETG, while it is associated with higher bond values on more elastic substrates such as TPU and PCTG. In the special case of tri-layer aligners, identifying the outermost layer is critical to select a compatible bonding workflow [30]. For example, SmartTrack (Align Technology, San Jose, Calif) is arranged PU-PETG-PU, whereas ClearQuartz (ClearCorrect LLC, Round Rock, Tex) is PETG-PU-PETG [31]. For 3D-printed aligners, a universal primer combined with an orthodontic adhesive shows the most consistent performance. Sandblasting improves adhesion for TA28 and DCA, whereas TC-85 performs slightly better without sandblasting. Additionally, the relatively high bond strengths observed in this study suggest that auxiliaries could be bonded directly to the tray rather than to enamel, allowing orthodontists to add elastic attachments, bite turbos, distalizing pads, or mandibular advancement hooks without cutting a precision window in the tray and without re-fabricating an entirely new series of aligners.

This in vitro study has inherent constraints. Substrates were prepared as flat plates rather than trays conforming to the dental arch, therefore, geometry-dependent stress distributions were not reproduced. Aging was limited to short thermocycling without prolonged water storage, cyclic mechanical fatigue, chemical challenges, or mastication simulation. Surface preparation was restricted to alumina sandblasting, excluding alternative methods such as laser treatment or other chemical primers. Only two bonding strategies and a limited set of commercial materials were evaluated, making the findings product- and substrate-specific rather than universally generalizable. Failure mode was scored with ARI at low magnification, without fractography to localize crack paths more precisely. Together, these factors indicate that bench-top strengths likely overestimate in vivo performance where aligner flexure and peel are present. In addition, localized sandblasting may leave residual abrasive particles or create surface thinning of the tray, which could affect comfort and tray integrity; biocompatibility and long-term effects of chairside micro-etching were not assessed and warrant further evaluation.

## 5. Conclusions

Bonding efficacy is primarily driven by the aligner substrate, with notable interactions among material, surface preparation, and primer system. Sandblasting generally increases shear bond strength through enhanced micromechanical retention, with some exceptions. In thermoformed trays, the one-step aligner adhesive tends to outperform the universal primer. For PETG, sandblasting reduces bond strength, whereas the more elastic TPU and PCTG show gains with sandblasting. In 3D-printed trays, the universal primer combined with an orthodontic adhesive performs more consistently. Sandblasting benefits DCA and TA-28, while TC-85DAC performs slightly better without it. Substrate shear strength can limit the adhesive system once a bond-strength threshold is surpassed. Clinically, lower effective strengths than in vitro values should be anticipated, and protocol selection is best tailored to the specific aligner material and expected loading conditions.

## Supporting information

**S1 File. Dataset.**
(XLSX)

## Author contributions

**Conceptualization:** Viet Anh Nguyen.

**Data curation:** Viet Anh Nguyen, Thi Quynh Trang Vuong.

**Formal analysis:** Viet Anh Nguyen, Thi Quynh Trang Vuong.

**Funding acquisition:** Viet Anh Nguyen, Viet Hoang.

**Investigation:** Viet Anh Nguyen, Thi Quynh Trang Vuong, Thi Nga Phung.

**Methodology:** Viet Anh Nguyen.

**Project administration:** Viet Anh Nguyen.

**Resources:** Viet Anh Nguyen, Viet Hoang, Nghi Phan Bich Hoang.

**Software:** Viet Anh Nguyen, Viet Hoang.

**Supervision:** Viet Anh Nguyen.

**Validation:** Viet Anh Nguyen, Thi Nga Phung.

**Visualization:** Viet Anh Nguyen.

**Writing – original draft:** Viet Anh Nguyen.

**Writing – review & editing:** Viet Anh Nguyen.

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
