## [Decision Letter · Decision Letter 0]

14 Sep 2025

Dear Dr. Nguyen,

Thank you for submitting your manuscript to PLOS ONE. After careful consideration, we feel that it has merit but does not fully meet PLOS ONE’s publication criteria as it currently stands. Therefore, we invite you to submit a revised version of the manuscript that addresses the points raised during the review process.

We look forward to receiving your revised manuscript.

Kind regards,

Rawaa A. Faris, Ph.D.

Academic Editor

PLOS ONE

Journal Requirements:

2. We note that Figure 1 in your submission contain copyrighted images. All PLOS content is published under the Creative Commons Attribution License (CC BY 4.0), which means that the manuscript, images, and Supporting Information files will be freely available online, and any third party is permitted to access, download, copy, distribute, and use these materials in any way, even commercially, with proper attribution. For more information, see our copyright guidelines: http://journals.plos.org/plosone/s/licenses-and-copyright.

Reviewers' comments:

Reviewer's Responses to Questions

**Comments to the Author**

1. Is the manuscript technically sound, and do the data support the conclusions?

Reviewer #1: Yes

Reviewer #2: Yes

2. Has the statistical analysis been performed appropriately and rigorously?

Reviewer #1: No

Reviewer #2: I Don't Know

3. Have the authors made all data underlying the findings in their manuscript fully available?

Reviewer #1: Yes

Reviewer #2: No

4. Is the manuscript presented in an intelligible fashion and written in standard English?

Reviewer #1: Yes

Reviewer #2: No

Reviewer #1: The paper titled ‘ Bond strength of thermoformed and 3D-printed aligners with universal primer versus one-step aligner adhesive with and without sandblasting: an in vitro study” is of good scientific quality and well structured. However, some critical points need to be clarified and further discussion is required before acceptance. As follows:

1- The statement that “Published evidence is virtually confined to one study” require justification with more references. Were there truly no other related studies? Add more references to support your claim.

2- What is the clinical siginificance of direct bonding to alignes VS enamel?

3- In the methodology, there are some areas need clarifications

• Please provide a schematic diagram or flow chart of the sample preparation and grouping for clarification.

• Why only alumina sandblasting used? Explain the reason for selecting this type of surface treatment.

• Why 500 thermocycles were selected?

4- Results are comprehensive, but:

• table 4 is difficult to follow

• Bond strengths is very high. Any comments?

• “Failure mode was material-dependent and not completely parallel with bond strength”. Please explain.

• Figures (2-5) need further explaniations.

5- In the discussion, clinical translation could be expanded.

6- The language is good and overall clear but some paragraphs in the discussion need to be shorten.

7- Ensure consistent use of terminology: eg., Bond aligner or aligned adhesive.

Recommendation: Minor to moderate revision. The manuscript is strong and novel but require clarification in methodology, results and discussion.

Reviewer #2: Many Presentation Errors

Line 46: "thermocycles" → "thermocycles"

Line 96: "emphasising" → "emphasizing"

Line 182: "Commercia l name" → "Commercial name"

Line 185: "modulu s (Mpa)" → "modulus (MPa)"

Line 260: "photo-polymerised" → "photopolymerized"

Line 392: "noTable 1nteractions" → "notable interactions"

Line 520: "Thermo-forme d foils" → "Thermo-formed foils"

Line 670: "percent" → "per cent" (align with PLOS ONE style consistency)

Methodology:

Substrates were tested as flat plates rather than actual aligners, which may not replicate intraoral stresses and force distributions. Emphasize in Methods and Discussion that geometry-specific stress and flexure effects are not represented; recommend future studies on full-arch aligners. discuss these issues in the discussion in the limitation section

Artificial aging was limited to 500 thermal cycles, approximating only 1–2 weeks of clinical wear. Acknowledge that longer-term degradation (e.g., >10,000 cycles, water storage, pH cycling, or mastication simulation) would provide more realistic performance predictions.

Only one surface treatment (alumina sandblasting) was assessed. Recommend testing alternative methods such as laser etching, plasma treatment, or different primers to broaden clinical applicability.

Limited range of adhesives evaluated (one universal primer and one aligner-specific adhesive). Note that findings are adhesive-specific; suggest inclusion of additional commercial products for more generalizable conclusions.

Failure mode analysis was conducted at low magnification with ARI scoring only. Recommend fractographic analysis (e.g., SEM of fracture surfaces) to provide more precise insights into crack propagation and interfacial failure.

Discussion:

Clinical translation may be overstated, as in vivo aligner flexibility and moisture could significantly lower effective bond strength compared to rigid backing test conditions. Expand discussion to clarify that bench-top results likely overestimate clinical values; suggest in vivo validation.

Limited comparison with broader literature. Strengthen discussion by integrating findings from other orthodontic bonding studies (beyond aligners), highlighting common adhesive principles.

Lack of focus on clinical thresholds. Discuss whether observed bond strengths exceed clinically acceptable limits (e.g., 6–8 MPa) and what this means for auxiliary stability.

No mention of potential patient safety concerns (residual abrasive particles from sandblasting, risk of substrate damage). Add a note on clinical precautions and the need to evaluate biocompatibility and long-term effects of sandblasting on aligner integrity.

Limited acknowledgment of generalizability. Explicitly state that results are material- and product-specific, and may not extrapolate to other aligner brands or future formulations.

**Do you want your identity to be public for this peer review?** For information about this choice, including consent withdrawal, please see our Privacy Policy

Reviewer #1: No

Reviewer #2: No

---

## [Author Response · Author response to Decision Letter 1]

29 Oct 2025

October 29th, 2025

Dear Editorial Board, PLOS ONE,

We have revised the manuscript thoroughly according to the comments of the reviewers. Any revisions made in our manuscript document were highlighted in red. Please help us review the manuscript again.

Sincerely,

Reviewer #1: The paper titled ‘ Bond strength of thermoformed and 3D-printed aligners with universal primer versus one-step aligner adhesive with and without sandblasting: an in vitro study” is of good scientific quality and well structured. However, some critical points need to be clarified and further discussion is required before acceptance. As follows:

Comment: 1- The statement that “Published evidence is virtually confined to one study” require justification with more references. Were there truly no other related studies? Add more references to support your claim.

Answer: Thank you for your comment. We agree and have expanded the final paragraph of the Introduction:

“To our knowledge, published evidence on bonding to aligner plastics is virtually confined to several studies [8,9]. Pariyatdulapak et al. reported highly variable bond strengths when metal mesh buttons were bonded to PETG and TPU foils with one paste adhesive. However, the button mesh conferred additional mechanical retention, and the test method did not follow ISO 29022:2013, limiting the relevance of its findings to direct composite to aligner bonding [10]. Another technical report on attaching buttons directly onto clear aligners mainly describes a chairside method for creating elastic anchors, without reporting standardized shear bond strength or characterizing the composite-to-aligner interface [9].”

Comment: 2- What is the clinical siginificance of direct bonding to alignes VS enamel?

Answer: Thank you. The clinical significance of direct bonding to alginers have been added to the Introduction section:

“Direct bonding of hooks or buttons to the aligner itself has therefore gained popularity [3]. An in vitro force distribution study recently showed that bonded composite buttons transmitted roughly 96 % of the applied elastic load and dispersed it across several teeth, whereas laser-cut hook traction lost more force and focused stresses chiefly on the canines, emphasising the biomechanical advantage of directly bonding buttons to the aligner [4]. Beyond elastics, chairside bonding can facilitate anterior bite turbos or posterior bite blocks for open bite correction or vertical control, applications where bulky virtual resin ramp can leave a large unsupported void beneath the tray and thereby compromise its fit. Functional or orthopedic adjuncts are also conceivable, such as mandibular advancement devices or bondable pads of maxillary distalizers, which could be bonded directly to aligners [5].”

Comment: 3- In the methodology, there are some areas need clarifications

• Please provide a schematic diagram or flow chart of the sample preparation and grouping for clarification.

Answer: Thank you. A flow chart has been added:

“Fig. 1. Study flowchart. Each material was split into two surface-treatment conditions: no treatment and sandblasting. Within each surface condition, specimens were bonded using either a two-step universal primer-orthodontic adhesive combination or a one-step aligner adhesive. After thermocycling, all bonded specimens underwent shear bond strength testing.”

Comment: • Why only alumina sandblasting used? Explain the reason for selecting this type of surface treatment.

Answer: Thank you, the reason for the single surface treatment has been added:

“Sandblasting was selected as the sole surface treatment because it is widely applied in dental bonding, whereas alternative methods such as laser texturing, plasma, or chemical etchants are not routine for clear aligners and may not be approved for intraoral use on proprietary tray materials [12].”

Comment: • Why 500 thermocycles were selected?

Answer: Thank you, the reason for 500 thermocycles has been added:

“The cycle count was selected to approximate 1-2 weeks of intra oral service [13]. Because the widely accepted conversion proposed by Gale et al., about 10,000 cycles per clinical year, 500 cycles represent the thermal load experienced over the maximum recommended aligner wear interval of 14 days [14].”

Comment: 4- Results are comprehensive, but:

• table 4 is difficult to follow

Answer: Thank you. Table 4 is intended primarily as a numerical summary of each experimental cell (i.e., each unique combination of material × surface treatment × bonding protocol), reporting the mean ± SD shear bond strength in MPa and the statistical groupings. In other words, the table is meant to let the reader directly compare which specific protocol–material pairings are statistically different, which requires showing the post hoc rankings (superscript letters) alongside the raw values. For visual interpretation, we direct the reader to Figure 3, which presents the same data as box plots grouped by material and protocol.

Comment: • Bond strengths is very high. Any comments?

Answer: Thank you. A discussion on high bond strength have been added:

“The discrepancy likely reflects methodology, in which the earlier button–sheet joint wasn’t bonded to a rigid industrial-plastic backing. Therefore, the sheet bent under load during testing, facilitating peel and easier debonding, whereas our direct composite-to-aligner protocol, with rigid support, isolates true adhesive performance. Furthermore, the peak values are still within the general magnitude recently reported for composite attachments bonded to other materials, suggesting that the present results are biomechanically plausible rather than artefactually inflated [26].

Clinically, these findings should be interpreted with caution. Our specimens were supported by a rigid backing to suppress tray flexion, whereas aligners in vivo bend under load, introducing peel components that can reduce effective shear capacity. Therefore, although most groups exceeded the commonly cited clinical threshold of approximately 6-8 MPa, real-world performance may be lower [27,28].”

Comment: • “Failure mode was material-dependent and not completely parallel with bond strength”. Please explain.

Answer: Thank you, an explanation on the non-parallel between failure mode and bond strength has been added:

“Notably, the PETG-non-sandblasted-Bond Aligner combination showed the largest ARI 3 proportion and the highest bond strength, consistent with stronger adhesion at the aligner-adhesive interface. Across the remaining groups, the relation between bond strength and ARI was not monotonic, likely because failure mode also depends on the cohesive shear capacity of the aligner substrate. The predominance of mixed failures with tearing within the tray indicates that once interfacial adhesion surpasses the tray’s cohesive strength, fracture propagates through the polymer sheet rather than through the adhesive, yielding variable ARI distributions even at comparable bond strengths.”

Comment: • Figures (2-5) need further explaniations.

Answer: Thank you. Further explanations have been added to figures 2-5:

“Fig. 2. Scanning electron micrographs (×1000) of aligner surfaces before (left panel) and after sandblasting (right panel). A,B, PETG; C,D, TPU; E,F, PCTG; G,H, TA-28; I,J, TC-85DAC; K,L, DCA. Thermoformed foils and TA-28 showed pronounced ploughing, pits, and sharp asperities after sandblasting. TC-85DAC and DCA showed comparatively shallower micro-cracking and less crater formation.

Fig. 3. Box plots of shear bond strength (MPa) by material, surface treatment (non-sandblasted versus sandblasted), and primer (universal primer–orthodontic adhesive combination versus one-step aligner adhesive).

Fig. 4. Distribution of failure modes with Adhesive Remnant Index (ARI) across combinations of material, primer, and surface treatment. PETG bonded with Bond Aligner tended to show higher ARI 3, whereas PCTG showed mostly ARI 0.

Fig. 5. Representative failure modes as per Adhesive Remnant Index (ARI). A, ARI 0, adhesive fully on the composite cylinder; B, ARI 1, mixed failure with <50% adhesive left on tray; C, ARI 2, mixed failure with >50% adhesive left on tray; D, ARI 3, cohesive failure within the adhesive or tray polymer.”

Comment: 5- In the discussion, clinical translation could be expanded.

Answer: Thank you for your comment. Clinical translation have been expanded:

“From a clinical standpoint, the study results suggest a decision tree rather than a ‘one size fits all’ protocol. For thermoformed aligners, Bond Aligner is preferable to a universal primer. Sandblasting is contraindicated for PETG, while it is associated with higher bond values on more elastic substrates such as TPU and PCTG. In the special case of tri-layer aligners, identifying the outermost layer is critical to select a compatible bonding workflow [29]. For example, SmartTrack (Align Technology, San Jose, Calif) is arranged PU-PETG-PU, whereas ClearQuartz (ClearCorrect LLC, Round Rock, Tex) is PETG-PU-PETG. For 3D-printed aligners, a universal primer combined with an orthodontic adhesive shows the most consistent performance. Sandblasting improves adhesion for TA28 and DCA, whereas TC-85 performs slightly better without sandblasting. Additionally, the relatively high bond strengths observed in this study suggest that auxiliaries could be bonded directly to the tray rather than to enamel, allowing orthodontists to add elastic attachments, bite turbos, distalizing pads, or mandibular advancement hooks without cutting a precision window in the tray and without re-fabricating an entirely new series of aligners.”

Comment: 6- The language is good and overall clear but some paragraphs in the discussion need to be shorten.

Answer: Thank you for your comment. We have condensed the Discussion paragraphs and removed repetitions.

Comment: 7- Ensure consistent use of terminology: eg., Bond aligner or aligned adhesive.

Recommendation: Minor to moderate revision. The manuscript is strong and novel but require clarification in methodology, results and discussion.

Answer: We appreciate the comment on consistency. We carefully reviewed the manuscript and attempted to enforce a single uniform term for the one-step aligner-specific bonding protocol. However, using only one label throughout created local ambiguity in some sections, because in certain contexts we needed to distinguish between (i) the commercial one-step aligner adhesive system and (ii) the conventional two-step “universal primer + orthodontic adhesive” protocol. To balance clarity and avoid unnecessary repetition, we keep using those same two phrases in parallel.

Reviewer #2:

Comment: Many Presentation Errors

Line 46: "thermocycles" → "thermocycling"

Line 96: "emphasising" → "emphasizing"

Line 182: "Commercia l name" → "Commercial name"

Line 185: "modulu s (Mpa)" → "modulus (MPa)"

Line 260: "photo-polymerised" → "photopolymerized"

Line 392: "noTable 1nteractions" → "notable interactions"

Line 520: "Thermo-forme d foils" → "Thermo-formed foils"

Line 670: "percent" → "per cent" (align with PLOS ONE style consistency)

Answer: Thank you for your comment. Above typos have been corrected.

Comment: Methodology:

Substrates were tested as flat plates rather than actual aligners, which may not replicate intraoral stresses and force distributions. Emphasize in Methods and Discussion that geometry-specific stress and flexure effects are not represented; recommend future studies on full-arch aligners. discuss these issues in the discussion in the limitation section.

Answer: Thank you for your comment. The above limitation has been acknowledged.

“Substrates were prepared as flat plates rather than trays conforming to the dental arch, therefore, geometry-dependent stress distributions were not reproduced.”

Comment: Artificial aging was limited to 500 thermal cycles, approximating only 1–2 weeks of clinical wear. Acknowledge that longer-term degradation (e.g., >10,000 cycles, water storage, pH cycling, or mastication simulation) would provide more realistic performance predictions.

Answer: Thank you for your comment. The above limitation has been acknowledged.

“Aging was limited to short thermocycling without prolonged water storage, cyclic mechanical fatigue, or chemical challenges.”

Comment: Only one surface treatment (alumina sandblasting) was assessed. Recommend testing alternative methods such as laser etching, plasma treatment, or different primers to broaden clinical applicability.

Answer: Thank you for your comment. The above limitation has been acknowledged.

“Sandblasting was selected as the sole surface treatment because it is widely applied in dental bonding, whereas alternative methods such as laser texturing, plasma, or chemical etchants are not routine for clear aligners and may not be approved for intraoral use on proprietary tray materials [12].”

“Surface preparation was restricted to alumina sandblasting, excluding alternative methods such as laser treatment or other chemical primers.”

Comment: Limited range of adhesives evaluated (one universal primer and one aligner-specific adhesive). Note that findings are adhesive-specific; suggest inclusion of additional commercial products for more generalizable conclusions.

Answer: Thank you for your comment. The above limitation has been acknowledged.

“Only two bonding strategies and a limited set of commercial materials were evaluated, making the findings product- and substrate-specific rather than universally generalizable.”

Comment: Failure mode analysis was conducted at low magnification with ARI scoring only. Recommend fractographic analysis (e.g., SEM of fracture surfaces) to provide more precise insights into crack propagation and interfacial failure.

Answer: Thank you for your comment. The above limitation has been acknowledged.

“Failure mode was scored with ARI at low magnification, without fractography to localize crack paths more precisely.”

Comment: Discussion:

Clinical translation may be overstated, as in vivo aligner flexibility and moisture could significantly lower effective bond strength compared to rigid backing test conditions. Expand discussion to clarify that bench-top results likely overestimate clinical values; suggest in vivo validation.

Answer: Thank you. The above discussion has been expanded:

“Clinically, these findings should be interpreted with caution. Our specimens were supported by a rigid backing to suppress tray flexion, whereas aligners in vivo bend under load, introducing peel components that can reduce effective shear capacity. In addition, intraoral moisture and continuous water exposure may further weaken the interface over time, which was not replicated in the present setup. Therefore, although most groups exceeded the commonly cited clinical threshold of approximately 6-8 MPa, real-world performance may be lower, suggesting in vivo validation [27,28].”

Comment: Limited comparison with broader literature. Strengthen discussion by integrating findings from other orthodontic bonding studies (beyond aligners), highlighting common adhesive principles.

Answer: Thank you for your comment. More studies on orthodontic bonding have been added for comparisons:

“Compared with the sole prior report of Pariyatdulapak et al.[8], which bonded orthodontic buttons with mesh bases to aligner materials using Bond Aligner and found 4.63-7.04 MPa (PETG) and 1.17-1.40 MPa (TPU), with PETG failures dominated by substrate tear-out, we observed markedly higher strengths (26.71 MPa on PETG and 5.31 MPa on TPU). The discrepancy likely reflects methodology, in which the earlier button–sheet joint wasn’t bonded to a rigid industrial-plastic backing. Therefore, the sheet bent under load during testing, facilitating peel and easier debonding, whereas our direct composite-to-aligner protocol, with rigid support, isolates true adhesive performance. Furthermore, the peak values are still within the general magnitude recently reported for composite attachments bonded to other materials (12-15 MPa), suggesting that the present results are biomechanically plausible rather than artefactually inflated [26-28].”

Comment: Lack of focus on clinica

---

## [Decision Letter · Decision Letter 1]

11 Jan 2026

Bond strength of thermoformed and 3D-printed aligners with universal primer versus one-step aligner adhesive with and without sandblasting: an in vitro study

PONE-D-25-44356R1

Dear Dr. Viet Anh Nguyen

We’re pleased to inform you that your manuscript has been judged scientifically suitable for publication and will be formally accepted for publication once it meets all outstanding technical requirements.

Kind regards,

Rawaa A. Faris, Ph.D.

Academic Editor

PLOS One

Additional Editor Comments (optional):

Reviewers' comments:

Reviewer's Responses to Questions

**Comments to the Author**

Reviewer #1: All comments have been addressed

2. Is the manuscript technically sound, and do the data support the conclusions?

Reviewer #1: Yes

3. Has the statistical analysis been performed appropriately and rigorously?

Reviewer #1: Yes

4. Have the authors made all data underlying the findings in their manuscript fully available?

Reviewer #1: Yes

5. Is the manuscript presented in an intelligible fashion and written in standard English?

Reviewer #1: Yes

Reviewer #1: (No Response)

**Do you want your identity to be public for this peer review?** For information about this choice, including consent withdrawal, please see our Privacy Policy

Reviewer #1: **Yes:** RANEEN KH. AL-HAMD

---

## [Editor Report · Acceptance letter]

PONE-D-25-44356R1

PLOS One

Dear Dr. Nguyen,

I'm pleased to inform you that your manuscript has been deemed suitable for publication in PLOS One. Congratulations! Your manuscript is now being handed over to our production team.

Kind regards,

on behalf of

Dr. Rawaa A. Faris

Academic Editor

PLOS One